# The Natural History of Retinal Vascular Changes from Infancy to Adulthood in Wyburn-Mason Syndrome

**DOI:** 10.3390/medicina56110598

**Published:** 2020-11-09

**Authors:** Kristina Horkovicova, Ivajlo Popov, Dana Tomcikova, Veronika Popova, Vladimir Krasnik

**Affiliations:** 1Department of Ophthalmology, Faculty of Medicine, Comenius University, 821 01 Bratislava, Slovakia; k.horkovicova@gmail.com (K.H.); ivajlo.popov@gmail.com (I.P.); 2Department of Pediatric Ophthalmology, Faculty of Medicine, Comenius University, The National Institute of Children’s Diseases, 82101 Bratislava, Slovakia; dana.tomcikova@nudch.eu (D.T.); veronika.labuzova@gmail.com (V.P.)

**Keywords:** Wyburn-Mason syndrome, arteriovenous malformations, angiomatosis

## Abstract

Wyburn-Mason syndrome is a rare, non-hereditary congenital neurocutaneous disorder leading to arteriovenous malformations. Malformations are characterized by an artery that is directly connected to veins without a capillary system and forms a fragile mass of abnormal vessels. It can be found in the midbrain, in the eyes, orbit, and rarely in cutaneous nevi. Neurological and ocular symptoms are the most common. Ocular signs and symptoms include abnormally dilatated vessels of conjunctiva, nystagmus, strabismus, vitreous hemorrhage, vein occlusions, retinal detachment, etc. Neurological symptoms may include headaches, paralysis, epistaxis, hydrocephalus, and hemiparesis. Imaging modalities such as MRI/CT angiography, optical coherence angiography, and fluorescein angiography are the most important for the identification of arteriovenous malformations. In our case report, we present an eight-month-old girl with an incidental finding of retinal angiomatosis on the left eye and was subsequently diagnosed with Wyburn-Mason syndrome. We compare the findings from the first visit to her clinical findings 20 years later.

## 1. Introduction

Retinal racemose hemangioma is a benign non-hereditary arteriovenous communication that may occur as an isolated lesion or in the setting of Wyburn-Mason syndrome (WMS). The first case of WMS was documented in 1874 as an ophthalmological curiosity [1]. In 1932, Yates and Payne again described an arteriovenous malformation in the retina and brain [2]. In 1937, Bonnet, Dechaume, and Blanc published two cases with similar malformations [3]. This anomaly was named after Wyburn-Mason who collected all the published cases in 1943 [4]. Malformations mostly occur in the retina and brain and rarely on the facial skin. Unlike other phacomatosis with retinal involvement, WMS rarely affects the skin [5,6]. It has three main types. In type I, the malformations are characterized by abnormal arterio-venous communication. Type II lesions have no capillary bed, and in type III lesions, it is not possible to distinguish the arteries from veins [7]. No exact data about the prevalence in the population exist [8]. The hereditary pattern, sex, or race predilection is not known. The exact etiology or risk factors are not clear yet. The unilateral involvement is the most common, although a bilateral presentation has also been reported [8]. These lesions are direct artery-to-vein communications that cause high arterial blood flow in the malformations. Vessels with high blood flow are susceptible to turbulent flow and vessel wall damage, which can lead to thrombosis and occlusion [9]. Wyburn-Mason syndrome is caused by an anomaly in organogenesis. No familial forms have been confirmed yet. Cells in the vascular walls of these regions originate embryologically from the cephalic regions. The cells of vessels in the facial, orbital, maxillary or mandibular, and encephalic areas come from three large embryonic regions. Disturbance in the migration of these cells causes vascular malformations known as cerebrofacial. These malformances are known as cerebrofacial metameric or segmental syndromes (CAMS) [8]. They are called CAMS1, CAMS2, and CAMS3 according to the origin of the cells (CAMS1: Corpus callosum, hypothalamus, olfactory tract, forehead, nose; Wyburn-Mason syndrome that has been renamed CAMS2: Cortex and diencephalon, optic chiasma, optic nerve, retina, sphenoid, maxilla, cheek; CAMS3: Cerebellum, temporal bone, mandible).

This case report documents the presentation of a child diagnosed in infancy with Wyburn-Mason syndrome and discusses her clinical follow-up 20 years later. It also describes the AV malformation changes after 20 years in the same patient.

## 2. Case Presentation

The patient was born at 35 weeks of gestation (in 1988) and at 8-months of age, was incidentally found to have retinal angiomatosis of her left eye. Due to a misdiagnosis with retinopathy od prematurity, she underwent periphrenal retinocryopexy at another institution. A fundoscopis examination (Figure 1A) performed under general anesthesia showed no abnormalities in the right eye. In the left eye, the optic disc had blurred boarders and was covered with markedly tortuous retinal vessels that were prominent into the vitreous. The dilation of the vessels gradually faded into the periphery. The macula was edematous and without foveal reflex. Temporally, there were scars after the previous cryopexy. Fluorescein angiography (Figure 2A–C) showed dilated vessels with no dye leakage in the macula region in all phases and a discrete leakage of dye from the vessels around the optic disc. Computed tomography angiography of the brain and orbits showed no abnormalities except for a stronger enhancement of the left eye’s retina. No treatment was suggested, a multidisciplinary follow-up was recommended. The patient was subsequently lost to follow-up.

In 2020, at the age of 21, the patient returned to our clinic for re-evaluation. The patient was being followed by a neurologist for management of headaches. She was also being followed by a pulmonologist and a hemangioma of the lung had been identified during screening for additional arteriovenous malformations. She was otherwise healthy.

According to the patient, the visual acuity had been stable for years. The best corrected visual acuity of the right eye was 20/20 and of the affected left eye was 20/125. The intraocular pressure was normal. The fundus examination of the right eye was unremarkable. The left eye (Figure 1B) optic disc showed blurred boarders, with thick and tortuous vessels running out from the optic disc. In the macular region, we found obliterated tortuous vessels. Temporally, at the mid-periphery, scars after cryopexy were visible. The retina was completely attached. Fluorescein angiography (Figure 2C–E) of the left eye showed dye leakage around the optic disc and few leaking points in the macula region in mid (Figure 2D) and late phases (Figure 2E). Diffuse leakage around the macula was observed in late phases (Figure 2E). No blood flow was visible in the obliterated vessels. Optical coherence tomography (OCT) (Figure 3) of the left eye showed retinal thinning with no clearly distinguishable inner retinal layers. OCT of the right retina was normal and no treatment was recommended.

## 3. Discussion

These congenital unilateral arteriovenous malformations may be of different sizes and locations [10]. Most often they involve the visual pathways from the retina to the occipital cortex, chiasm, hypothalamus, basal ganglia, midbrain, and cerebellum [11]. Malformations may grow, cause hemorrhages, thrombose, and involute with aging [9]. In differential diagnosis of retinal hemangiomas Von Hippel Lindau syndrome, vasoproliferative tumors, Sturge Weber syndrome, and retinal cavernous hemangioma should be excluded [12].

Larger malformations could affect vision to a different extent depending on their localization and size. Small arteriovenous malformations in the retina or visual pathway have a very small impact on visual function. Visual acuity could be affected by a direct compression of the visual centers or optic nerve head, choroidal infarctions, retina ischemia, or retinal vascular occlusions. None of the above-mentioned causes of visual loss were noted in our patient. High-flow malformations usually do not bleed or cause exudation in the retina [12]. No retinal edema, fluid accumulation, or bleeding was observed in our patient, but spots of clinically insignificant exudation were seen in the macular region on fluorescein angiography (Figure 2F). OCT shows changes in retinal thickness and organization of the layers. Changes could be the result of resorbed retinal edema or exudation in the past. No prior OCT scans of our patient are available to prove or reject this hypothesis.

Archer et al. classified retinal arteriovenous malformations into three groups. Group 1 is characterized by an abnormal capillary network, small arteriole-venule anastomoses which are difficult to detect clinically. Group 2 consists of direct AV communications which exclude the capillary network. AV connections have a marked hyperdynamic flow. Vision may be affected. In group 3, vessels are markedly convoluted, dilated, and tortuous. It is hard to distinguish arteries from veins. Vision loss is usually severe. Patients are most often diagnosed in childhood with a high risk of systemic vascular involvement [13].

The patient in this case is categorized as having group 3 anomalies. No bleeding, exudation, or edema was seen in adulthood. However, according to documentation, macular edema was present at the time of diagnosis in childhood, while undergoing routine screening for retinopathy of prematurity. Arteriovenous malformations in the lungs were diagnosed later in childhood but caused no symptoms. Headaches in patients with WMS are usually associated with AV malformations in the brain, but the patient in our case did not have any intracranial abnormalities and her headaches were unlikely to be related to her MWS diagnosis [14]. In Wyburn-Mason syndrome, a spontaneous involution of retinal and intracranial arteriovenous malformation was described [15,16]. This is in conformity with our case where Figure 1 and Figure 2 show a sclerotic vessel around the macula with no visible blood perfusion, which was proved in fluorescein angiography. Figure 1 and Figure 2 show changes of retinal vasculature from childhood to adulthood.

To our knowledge, this is the first published case documenting the natural history of retinal vascular changes from infancy to adulthood in MWS. There are two published cases of the evolution of these retinal lesions with a longer follow-up of 10 and 27 years, but these were reported in adults [17,18].

In conclusion, this report documents retinal vascular changes from infancy to adulthood in MWS using color fundus and fluorescein photography. Macular edema due to high blood flow perfusion was self-limited by vessel sclerotization. However, the retinal damage caused was permanent. Treatment of macular edema in infancy could prevent retinal damage, but treatment options are limited and current recommendations on management do not exist.

## Figures and Tables

**Figure 1 medicina-56-00598-f001:**
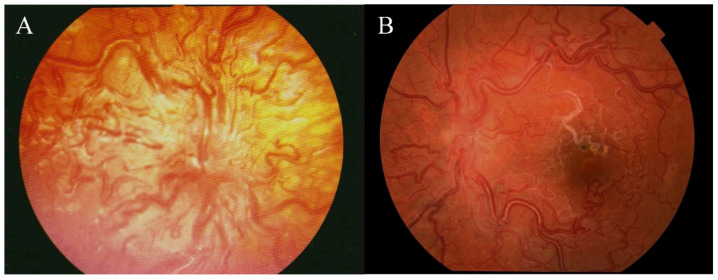
Color photography of the left eye fundus in (**A**) childhood, (**B**) adulthood.

**Figure 2 medicina-56-00598-f002:**
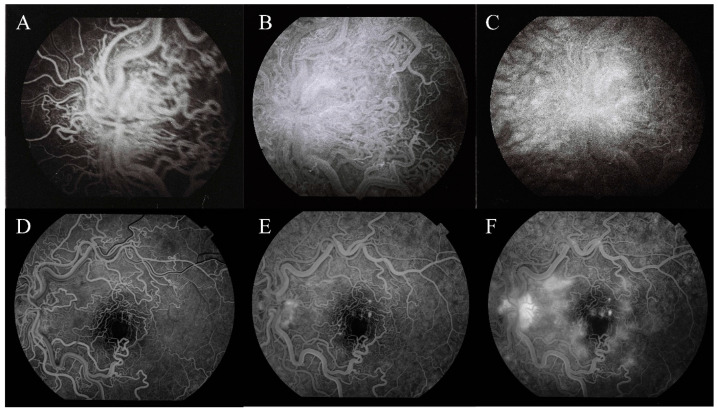
Fluorescein angiography of the left macular region. Childhood: Early (**A**), mid (**B**), and late (**C**) phases. Adulthood: Early (**D**), mid (**E**), and late (**F**) phases.

**Figure 3 medicina-56-00598-f003:**
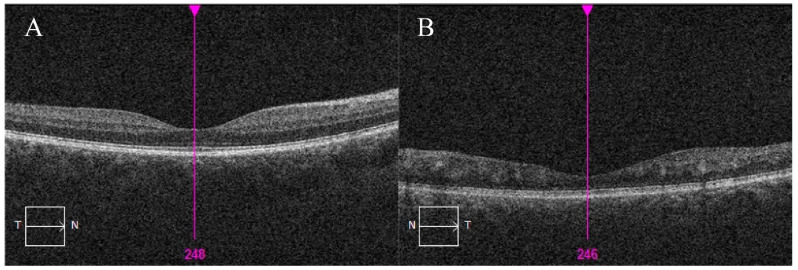
Optical coherence tomography (OCT) images of the right (**A**) and left (**B**) eye.

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
