# Peer review of "The Natural History of Retinal Vascular Changes from Infancy to Adulthood in Wyburn-Mason Syndrome"

_medicina, 2020, doi:10.3390/medicina56110598_

Round 1

Reviewer 1 Report

Dear Author(s),

Thanks for submitting your case report on Medicina Journal. Wyburn-Mason Syndrome (WMS) is a rare neuro-ocular syndrome still lacking of precise understanding of pathophysiology, risk factors and best treatment. From this point of view, your report is an admirable example of long-term follow-up of a typical ocular involvement case of WMS. The uniqueness derives from the possibility to review color retinographies and fluorescein angiography scans from baseline to 20-yrs follow-up. I have only minor remarks before publication:

Abstract

Pag1, line22: please revise the sentence in "In our case, we present a eight-month-old newborn...";

Introduction

Line28: please correct in "non-hereditary";

Line 43: I would delete the sentence "Etiology of this syndrome is not known today". You have spoken about the same subject in line 39;

Pag2, line 45: please correct in "origin";

Line 47: please revise as follow "Disturbance in the migration...";

In general, the introduction paragraph appears weighed down by too information. I would suggest to make this paragraph leaner. 

Case presentation

Line 66: please delete with, it is "Computed tomography angiography";

Line 70: please find a synonym for "follow", you have already used the same verb in the previous sentence;

Line 77: please revise in "mid-periphery";

Discussion

Pag3, line 93: this is the same historical fact you cited in the introduction. I would not use it another time;

Pag4, line 97: please revise in "Malformations may grow, cause hemorrhages, thrombose...".

Author Response

Dear Reviewer 1,

     we have made the correction according to your suggestions. Please see the attachment.

We hope you will find our remakes satisfying.

Best regards

Ivajlo Popov

Reviewer 2 Report

This case report is of interest and should be published as the only report in the literature documenting the natural history of retinal vascular changes in Wyburn-Mason syndrome from infancy to adulthood. 

The authors should be aware that an 8-month-old child is not considered to be a "newborn." This reference is used multiple times in the case report. 

The case report contains a significant number of English grammatical errors which I have mostly corrected. Please refer to the attached document. 

Author Response

Dear Reviewer 2,

     we have made the correction according to your suggestions. Please see the attachment.

We hope you will find our remakes satisfying.

Best regards

Ivajlo Popov
